# Visual Backtracking Teleoperation: A Data Collection Protocol for Offline Image-Based RL

**David Brandfonbrener**[*]  **Stephen Tu**[†]  **Avi Singh**[†]  **Stefan Welker**[†]

**Chad Boodoo**[†]  **Nikolai Matni**[†,‡]  **Jake Varley**[†]

## Abstract

We consider how to most efficiently leverage teleoperator time to collect data for learning robust image-based value functions and policies for sparse reward robotic tasks. To accomplish this goal, we modify the process of data collection to include more than just successful demonstrations of the desired task. Instead we develop a novel protocol that we call Visual Backtracking Teleoperation (VBT), which deliberately collects a dataset of visually similar failures, recoveries, and successes. VBT data collection is particularly useful for efficiently learning accurate value functions from small datasets of image-based observations. We demonstrate VBT on a real robot to perform continuous control from image observations for the deformable manipulation task of T-shirt grasping. We find that by adjusting the data collection process we improve the quality of both the learned value functions and policies over a variety of baseline methods for data collection. Specifically, we find that offline reinforcement learning on VBT data outperforms standard behavior cloning on successful demonstration data by 13% when both methods are given equal-sized datasets of 60 minutes of data from the real robot.

## 1 Introduction

A common approach to control from images is to collect demonstrations of task success and train a behavioral cloning (BC) agent [1, 2, 3]. This can lead to policies that are able to succeed on many tasks, particularly when mistakes do not cause the policy to move too far from the distribution of state-action transitions seen in the dataset of successful demonstrations. But, for many tasks (like T-shirt grasping) a mistake will take the policy out of this distribution and then the learned BC policy will fail to recover [4].

Such failures occur in part because a dataset of only successes does not contain enough information to recognize failures or learn recovery behaviors. To remedy this issue, we propose a novel data collection method called Visual Backtracking

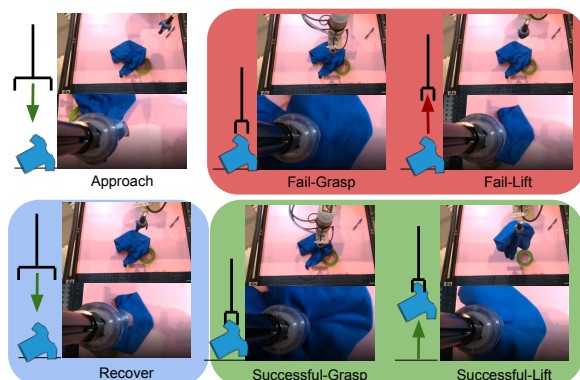

Figure 1: An illustration of our VBT method on a T-shirt grasping task. The method asks the teleoperator to demonstrate failure (red), recovery (blue), and success (green) within each trajectory. This provides the necessary coverage of failure and recovery to learn accurate value functions and robust policies while also preventing overfitting by ensuring that failures and successes are visually similar except for the task-relevant details.

---

[*]New York University, [†]Robotics at Google, [‡]University of Pennsylvania

Offline Reinforcement Learning Workshop at Neural Information Processing Systems, 2022

Teleoperation (VBT). Specifically, VBT leverages the teleoperator to collect visually similar failures, recoveries, and successes (Fig. 1).

VBT data collection is designed to combine well with offline reinforcement learning (OffRL) rather than BC. VBT data contains the necessary coverage of the state action space to learn accurate value functions as well as recovery behaviors. Then OffRL can leverage sparse rewards to automatically emulate the advantageous actions (including recovery behaviors) while avoiding the sub-optimal actions that led to the initial task failure.

It is crucial to the VBT method that the failures, recoveries, and successes are visually similar. Without this visual similarity, the OffRL learner can overfit to non-task-relevant elements of the image such as background clutter, leading to useless value functions. To easily maintain visual similarity, VBT collects each of failure, recovery, and success *within* the same trajectory (Fig. 1). Thus, VBT avoids overfitting, even on small, image-based datasets, by ensuring that differences between observations of failure and success are task-relevant.

Concretely, our contributions are:

1. We propose the novel VBT protocol for data collection to leverage human teleoperation to collect image-based datasets for use with OffRL. VBT resolves the two main issues with naive methods: (a) lack of coverage of failures and recoveries and (b) overfitting caused by visually dissimilar failures and successes.

2. We discuss how and why VBT can enable better policy learning via learning more accurate $Q$ functions and present empirical evidence of the improvement in $Q$ functions learned from VBT data compared to several baselines for data collection.

3. We present real robot results on a deformable grasping task to demonstrate the effectiveness of VBT data. When training from scratch on just one hour of robot time for data collection and image-based observations, a policy trained with OffRL on VBT data succeeds 79% of the time while BC trained on successful demonstrations has a success rate of 66%.

A detailed discussion of related work can be found in Appendix A, and a detailed discussion of the motivation behind our method can be found in Appendix B. For conciseness, we will refer to the estimates of $Q^\pi$ and $V^\pi$ learned by the critic of an OffRL algorithm like IQL [5] as $Q$ and $V$.

## 2   Our Method: Visual Backtracking Teleoperation (VBT)

Our main contribution is a novel method for data collection called Visual Backtracking Teleoperation (VBT). This method is particularly suited to collecting small image based datasets. The main idea is that in order to learn useful $Q$ functions and robust policies, we need a dataset that contains failures and recoveries as well as successes. Moreover, when learning from visual inputs we need the failures and recoveries to be as visually similar to the successes as possible to prevent overfitting and to encourage $Q$ functions and policies to use only the task-relevant information in the images.

Explicitly, VBT data collection consists of three steps for the teleoperator *within* each trajectory:

1. *Failure:* the teleoperator first fails at the desired task.
2. *Recovery:* the teleoperator recovers from the failure and begins to attempt the task again.
3. *Success:* the teleoperator successfully finishes the task.

The data is labeled with a sparse reward of 1 for the final transition and a small penalty otherwise.

VBT is a general recipe that can be applied to any sparse-reward task that has well-defined failure modes and is amenable to teleoperation. The failure modes need not be unique and the data could consist of several different types of failures. For our example task of deformable grasping these steps can be implemented as depicted in Fig. 1.

Compared to a dataset that contains only demonstrations of success, VBT contains better coverage of failure and recovery behaviors. As explained in Appendix B, this coverage is necessary to learn robust policies capable of recovery and $Q$ functions that can recognize failure.

Compared to a dataset that naively mixes failures and successes that are collected separately, VBT has the benefit of visually similar failures and successes. As explained in Appendix B, this prevents

the learned $Q$ functions and policies from overfitting based on visual details that are not task-relevant and instead forces the models to learn the task-relevant features that can generalize better.

# 3   Experimental Setup: baselines

The full details about our experimental setup can be found in Appendix C, here we define the baselines for our experiments. To understand the performance of VBT, we collect several different baseline datasets of robotic T-shirt grasping by human teleoperators. Each dataset contains 60 minutes of data from the real robot which equates to approximately $10,000$ steps in the environment.

**Success demonstrations (Success)**: The success dataset consists of demonstrations of task success. These episodes demonstrate efficient successful executions of the task with minimal recovery behaviors collected by a human teleoperator instructed to complete the task successfully.

**Success mixed with failures and recoveries (Coverage+Success)**: This dataset is a mixture of two datasets containing 30 minutes of robot data from each one. The first is the Success dataset described above. The second is a dataset containing repeated failures and recoveries and intended to provide coverage of task-relevant failure and recovery behaviors. Importantly, the coverage and success trajectories are collected independently.

**Success mixed with learning from play (LfP+Success)**: This dataset is also a mixture of two datasets containing 30 minutes of robot data from each one. The first is again the Success dataset. The second is a learning from play [6] dataset which contains data of the robot demonstrating rich interactions such as rolling lifting, dragging, and pushing, between the robot arm and a variety of different objects beyond the T-shirt.

**Visual Backtracking Teleoperation (VBT-Ours)**: Each episode in this dataset demonstrates failure, then recovery, and then task success as described in Section 2.

# 4   Experimental Results

We now present experimental results on our deformable grasping task that attempt to answer:

1. Are $Q$ functions trained on VBT data more accurate than those trained on other datasets?
2. Are policies trained on VBT data more successful than those trained on other datasets?

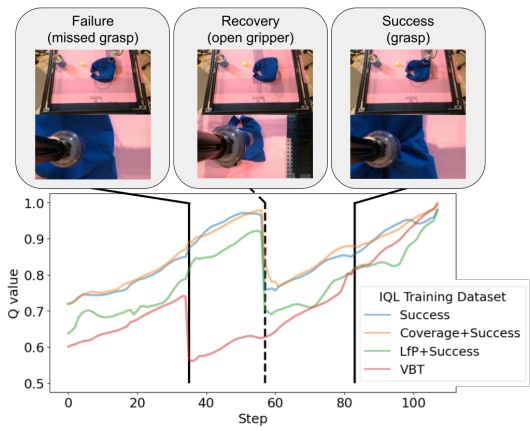

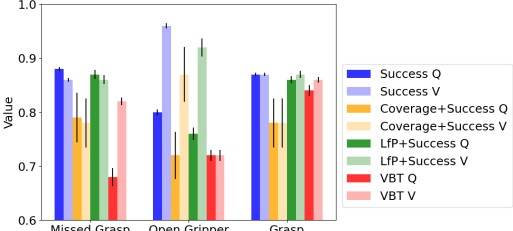

(b) Visualization of the average $Q$ and $V$ values at each of the three key steps highlighted in Fig. 2a across a held-out test set of 35 trajectories that are all similar to the one in subfigure (a). Error bars show standard error. Note that VBT has much lower $Q$ than $V$ for the missed grasp, while the baseline datasets have much lower $Q$ than $V$ for the open gripper, meaning the VBT value functions are more accurate.

(a) Evaluation of various $Q$ functions on a single held out evaluation trajectory that is not seen during training. The trajectory consists of a failure followed by recovery and a successful grasp. Notice that the VBT-trained $Q$ function is the only one to drop precisely at the point of the missed grasp instead of the open gripper action that is necessary for recovery.

Figure 2: Visualizations of $Q$ functions trainde with IQL on various datasets.

### 4.1 VBT data leads to improved value functions

To see the improvement in $Q$ functions when training on VBT data we present examples of the learned $Q$ functions on representative trajectories. From these examples we show that training $Q$ functions on VBT data resolves each of the issues raised in Section B. Namely, the $Q$ functions trained on VBT data (1) correctly identify failures while those trained on Success data do not and (2) resolve the overfitting issues that can arise when using Coverage+Success data.

First, we will show how $Q$ functions trained on VBT data correctly identify failures and recoveries. To do this we illustrate the $Q$ functions trained on Success, Coverage+Success, LfP+Success and VBT data on a single trajectory that contains failure, recovery, and success in Fig. 2a. As explained in Section B, the Success dataset does not provide sufficient information to understand failures. As a result, the $Q$ function trained on Success does not use the image observation to recognize failure. Instead it uses the proprioceptive state to correlates "gripper closed and moving up" with higher values *regardless of whether the shirt is in hand*. Training on VBT data resolves this issue by forcing the $Q$ function to attend to the information in the image to recognize when a failure has occurred. Thus, the $Q$ function trained on VBT drops precisely when the missed grasp happens.

To ensure that these results generalize beyond this single trajectory, we compute averages of the relevant value functions across a held-out test of VBT-style trajectories. The results are shown in Fig. 2b. The key observations are that as in Fig. 2a, the value functions trained on VBT correctly identify that the $Q$ value of the missed grasp is substantially lower than the $V$ value of the state from which the grasp occurs. This means that the OffRL algorithms using these value functions will correctly learn to avoid missed grasps. In contrast, training on any of the baseline datasets leads to $Q$ values that are above the $V$ value for the missed grasp (meaning they will learn to miss). And similarly, training IQL on the baseline datasets yields $Q$ values for the recovery behavior that are substantially below the $V$ values while training on VBT does not.

An evaluation of the overfitting problems when using Coverage+Success is in Appendix D.

### 4.2 VBT data leads to improved policies

To measure the impact of switching to VBT data on policy performance we perform an AB test. Specifically, we train policies using each of the three learning algorithms (BC, AWAC, IQL) on each of the four types of datasets (Success, Coverage+Success, LfP+Success, VBT-Ours), but excluding BC on the datasets that contain episodes that do not terminate in success (Coverage+Success, LfP+Success). For the AB test, each episode a policy is chosen at random and executed until either the policy sends the "terminate" action or the maximum number of steps per episode is reached. Success is determined by the majority vote out of three human labelers based on the final image in the episode. The randomness of the AB test ensures a valid comparison between all of the trained policies. Results are reported in Table 1.

There are several takeaways from these results:

| Dataset | Policy | Task Success |
|---|---|---|
| Success | BC | $66 \pm 4\%$ |
| Success | AWAC | $67 \pm 3\%$ |
| Success | IQL | $69 \pm 3\%$ |
| Coverage+Success | AWAC | $52 \pm 4\%$ |
| Coverage+Success | IQL | $64 \pm 3\%$ |
| LfP+Success | AWAC | $62 \pm 4\%$ |
| LfP+Success | IQL | $58 \pm 4\%$ |
| VBT-Ours | BC | $73 \pm 3\%$ |
| VBT-Ours | AWAC | $73 \pm 3\%$ |
| **VBT-Ours** | **IQL** | $\mathbf{79 \pm 3\%}$ |

Table 1: Results of an AB test of 1437 total episodes. Error bars report standard error.

1. Training IQL on VBT outperforms standard BC on Success data by 13% and outperforms IQL on Success data by 10%. These are significant gains, especially given that we are training each policy from scratch on very small datasets with image-based observations.

2. Due to overfitting, OffRL algorithms trained on Coverage+Success and LfP+Success underperform those same algorithms trained on Success alone.

3. Even BC on VBT data outperforms BC on Successes for this task since training BC on VBT leads to retrying behaviors. OffRL provides further benefits beyond BC by effectively filtering out the suboptimal actions that are present in VBT data.

The discussion section is deferred to Appendix E

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

## A  Related Work

Our work falls into the broader category of learning from demonstrations [7]. But, rather than taking a more traditional approach of BC from demonstrations of success [1, 2, 3, 8], we propose a novel method for collecting demonstrations of failure and recovery as well as success. The inclusion of failures in our dataset leads us to use OffRL [9, 10] to ensure that we do not imitate the failures. Specifically, we use the IQL [5] and AWAC [11] algorithms for OffRL. Some theoretical motivation for the use of OffRL rather than BC when learning from suboptimal data can be found in [12].

The insight that observations of failures is useful for policy learning has been made before in work on using success detectors to compute rewards in RL [13]. In contrast, VBT does not attempt to learn an explicit success detector, but provides a mechanism for collecting data that is especially useful for learning policies by capturing the salient differences between failure and success in visually similar observations.

While we consider a setting where training happens entirely offline (i.e., the data collection step is separated from the policy learning), there is a related line of work that collects examples of failures and recoveries by moving to an interactive setting where actions from partially trained policies are executed on the robot and judged by the human teleoperator. In particular the DAgger line of work exemplifies this pattern [4, 14, 15, 16, 17]. In contrast, VBT operates completely offline and does not require the teleoperator to interact with the learning process or to deploy learned policies on the robot during training.

VBT also is particularly suited to efficiently learn image-based value functions for sparse reward tasks. This capability could be especially useful in architectures like SayCan [2] that require image-based affordances for a variety of manipulation tasks, and it is an interesting direction for future work to use VBT as a component in training such larger systems.

## B  Motivation

VBT is designed to solve two issues that arise from simpler data collection methods: (1) lack of coverage of failure and recovery behaviors, and (2) overfitting issues that arise in the low-data and high-dimensional observation setting. Here we describe both of these issues in more detail before explaining how VBT resolves them.

### B.1  Coverage of failure and recovery

Any offline learning algorithm will be fundamentally limited by the quality of the dataset. We can only expect the algorithm to reproduce the best behaviors in the dataset, not to reliably extrapolate beyond them. So, when collecting datasets for offline learning of value functions and policies, we want to ensure sufficient coverage of the relevant states and actions. We argue that for sparse reward robotics tasks this requires including failures and recoveries in the dataset.

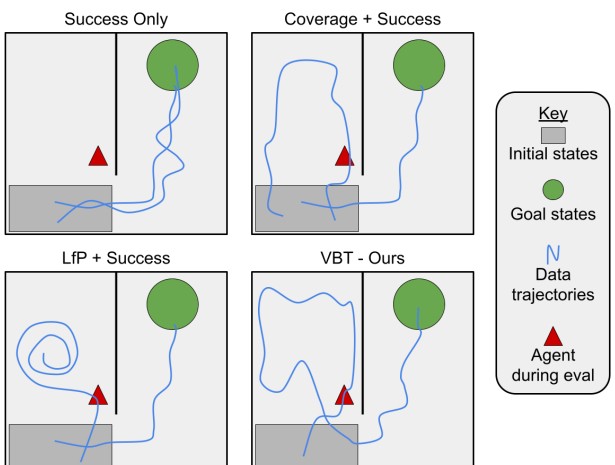

Figure 3: An illustration of each of the four types of datasets that we consider in a gridworld environment with a sparse reward for reaching the green goal. The Success Only dataset contains two successful trajectories. The Coverage+Success and LfP+Success datasets each contain one success and one failure. Our VBT dataset contains one trajectory that fails, then recovers, and then succeeds. Full descriptions of the datasets are in Section 3.

**Value functions require coverage.** Most OffRL algorithms involve estimating the $Q$ and $V$ value functions of a learned policy $\pi$ that is different from the policy (in our case the teleoperator) that collected the dataset. Explicitly, letting $\gamma \in [0, 1)$ be a discount factor and $r$ be the reward function they estimate $Q^\pi(s, a) = \mathbb{E}_\pi[\sum_{t=0}^\infty \gamma^t r(s_t, a_t) | s_0 = s, a_0 = a]$ and $V^\pi(s) = \mathbb{E}_{a \sim \pi|s}[Q^\pi(s, a)]$,

where expectations are taken over actions sampled from $\pi$. For the rest of the paper we will omit the superscript $\pi$ when clear from context.

The key issue with learning value functions for a policy $\pi$ that did not collect the data is that we can only reliably estimate value functions at states and actions that are similar to those seen in the training set [9]. This is born out by theoretical work which often requires strong coverage assumptions to learn accurate value functions, such as assuming that the data distribution covers all reachable states and actions [18].

While this sort of assumption is too strong to satisfy in practice, we argue that for the sparse reward robotic tasks that we consider, the relevant notion of coverage is to include failures and recoveries, as well as successes, in the dataset. These failures and recoveries can provide coverage of the task-relevant states and behaviors necessary to learn useful value functions. Without failures the learned value functions will not be able to identify the "decision boundary" between failure and success that is necessary for reliably accomplishing the specified task. Much as in supervised learning it is difficult to train a binary classifier without any examples of the negative class, we conjecture that it is difficult to learn accurate $Q$ functions for sparse reward tasks without any failures.

Consider the example datasets shown in Fig. 3. If we use the Success Only dataset to train a $Q$ function and then query the $Q$ function at the location of the red agent during evaluation we will get inaccurate $Q$ values. The learned $Q$ function does not have any indication that the presence of the wall between the agent and the goal exists since there is no training data from the left side of the wall. In contrast, any of the other datasets that contain failure examples where the agent enters the room to the left of the wall will allow the learned $Q$ function to assign lower values to states left of the wall compared to those right of the wall.

**Policies require coverage.** Much as with value functions, learned policies should not be expected to produce behaviors that are not present in the training set. This can be seen by looking at the policy loss function used in OffRL algorithms like IQL [5] and AWAC [11] which take the following form (where $\mathcal{D}$ is the distribution that generates the dataset and $Q, V$ are the learned $Q$ and $V$ value functions):

$$L(\pi) = \mathbb{E}_{s,a \sim \mathcal{D}}[\exp(Q(s,a) - V(s)) \cdot (-\log \pi(a|s))] \tag{1}$$

Notice that this is simply the standard negative log likelihood loss that is used for BC, except that each term is weighted by the exponentiated advantage function. This loss leads to policies that imitate actions with high advantages and ignore actions with low advantages. The key takeaway is that OffRL can only produce policies that choose actions that are already covered by the data distribution. So, if we want a robust policy that can recover from failures, we need to see recovery actions in the training set.

Note the DAgger line of work [4] raises a similar issue and resolves it by introducing online learning where the policy $\pi$ is executed on the system to expand coverage. Instead, we are considering an entirely offline setting where we want to produce the necessary coverage of failure states and recovery behavior a priori from teleoperation.

To see an example of why coverage of failures and recoveries is necessary for robust policy learning, again consider the datasets shown in Fig. 3. Imagine that at evaluation time, the agent reaches the red triangle. If the agent was trained on the Success Only dataset, it must rely on extrapolation to select an action, but there are never any examples of "down" in the dataset. As a result, the agent is stuck in the room to the left of the wall and cannot recover. In contrast, the other datasets may allow for the agent to learn a useful recovery behavior since they have better coverage.

### B.2 Overfitting in the low-data and image observation setting

When collecting data via teleoperation we are usually in the low-data regime since we are bottle-necked by teleoperator time on the real robot. Moreover, in the tasks that we consider here, the observation space is image-based and thus very high-dimensional. This combination of low-data and high-dimensional observations makes overfitting likely.

As explained above, in sparse reward tasks it is important for the $Q$ functions and policies to accurately represent the "decision boundary" between failure and success in the task. In simple observation spaces like the gridworld in Fig. 3, this may be fairly easy, e.g., as in identifying a failure by recognizing that the $(x, y)$ coordinated of the position are in the room to the left of the wall.

However, in high-dimensional image-based observation spaces that are encountered in real robotic tasks, this can become much more challenging.

Fig. 4 illustrates the challenges of image observations in our grasping task. If successes and failures are collected naively in separate episodes (Dataset A), there will be many visual differences between success and failure beyond the salient details about the gripper and shirt. For example, background clutter like the presence of a pen or the exact configuration of the folds of the shirt can spuriously correlate with failure. In contrast, Dataset B reduces the chances of overfitting by eliminating the spurious correlations. Separating the examples in Dataset B requires learning a model that is attuned to the subtle task-relevant visual differences between failure and success that can generalize well. Our method aims to collect data like Dataset B.

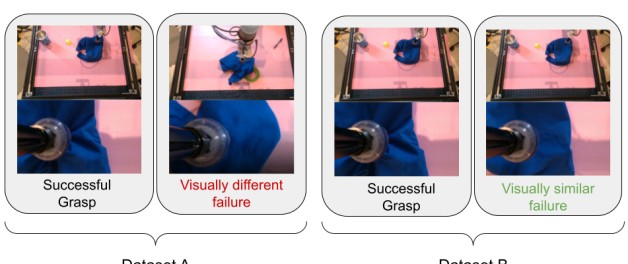

Figure 4: Datasets that include visually dissimilar successes and failures like Dataset A can cause overfitting. For example, a learned $Q$ function could learn to attend background distractions (like the pen) instead of the task-relevant details. In contrast, in Dataset B the only differences between success and failure are task-relevant.

## C  Experimental details

### C.1  Robot and Environment

Our setup consists of a reach-enabled [19] UR5 arm with a pneumatic powered 3-fingered gripper. We control the robot with a 5 dimensional action space: displacements in $(x, y, z)$ no larger than 5cm, a gripper toggle to either an open or closed position, and a termination indicator that immediately terminates an episode. The orientation of the end-effector is fixed. If the agent does not select the terminate action within 100 steps, we automatically terminate the episode.

The workspace consists of the same blue T-shirt, with various distractor objects and variations in lighting. The robot observation space consists of 2 camera images (360x640x3) from a wrist camera and overhead camera, as well as the Cartesian position of the end-effector $(x, y, z)$ and an indicator of whether the gripper is currently open or closed. We stack a sequence of 4 consecutive timesteps of observations as the input for our learning algorithms.

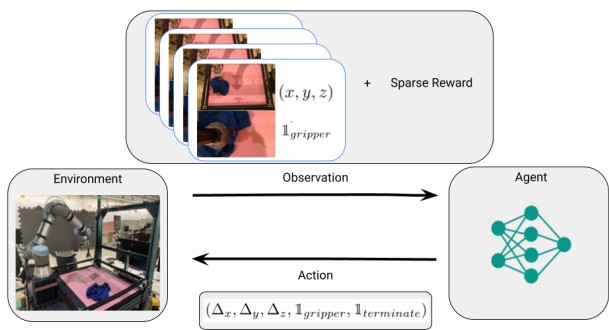

Figure 5: A description of the observation and actions spaces we use for our real robot experiments. The agent receives an observation of 4 timesteps of images, cartesian coordinates of the end effector, gripper status, and reward from the robot and sends back a 5 dimensional action of cartesian displacements and gripper and terminate commands.

The task is to lift the T-shirt more than 80% off a table. During training, as explained in Section 2, we automatically label each successful trajectory collected by the teleoperator with a terminal reward of 1. We give all other transitions a reward of -0.01 to encourage faster completion of the task. This environment is illustrated in Fig. 5.

## C.2 Learning algorithms

We run several different policy learning algorithms to understand utility of our datasets. **Behaviour Cloning (BC)** where policy is trained directly to imitate the actions in the dataset. **AWAC** [11] where an advantaged weighted actor-critic method is used to train both a policy $\pi$ and critic able to evaluate $V^\pi$ and $Q^\pi$. **IQL** [5] where implicit Q-Learning is used to train both a policy $\pi$ and critics $V^\pi$ and $Q^\pi$.

All policies and value functions are parameterized by the same neural architecture and trained using `JAX` [20] and `Flax` [21]. The image inputs are passed through a ResNet [22] encoder and then the features are concatenated along with the Cartesian coordinates of the gripper and the gripper indicator (as well as the action for $Q$ functions). These features are then passed through a simple multi-layer perceptron to output either an action or value.

All policies and value functions are trained from 700K gradient steps using the Adam [23] optimizer. During training images are aggressively augmented using changes in brightness, sharpness, color, contrast, artificial shadows, rotation, and cropping. This allows us to train the encoders from scratch on relatively small datasets. We use an inverse temperature of 0.1 for both IQL and AWAC, and we use expectile 0.9 for IQL, following [5].

## D Examining overfitting

The prior results indicate that the Coverage+Success does not resolve the coverage issues that are preventing us from learning accurate $Q$ functions. This may be somewhat surprising since this dataset was designed to have good coverage. We claim that this issue is due to overfitting in our low-data and high-dimensional observation setting. Essentially, the $Q$ function trained on Coverage+Success data uses the image observation to classify the trajectory as either "success" or "failure". If the trajectory is classified as "success" then the $Q$ function behaves just like a $Q$ function trained only on Success and associates "gripper closed and moving up" with higher $Q$ values. However, if the trajectory is classified as "failure" then the $Q$ function trained on Coverage+Success data behaves differently and assigns uniformly lower $Q$ values to the trajectory. Some hint of this can be seen in Fig. 2b where the values learned on Coverage+Success have substantially higher variance caused by some of the trajectories being classified as failures (note that LfP+Success behaves

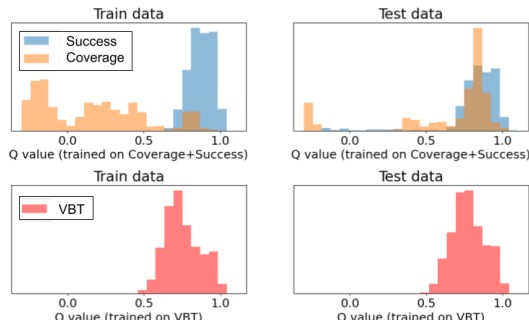

Figure 6: Histograms of the $Q$ values of IQL trained on Coverage+Success (Top) and VBT (Bottom) on both the data that the respective $Q$ functions were trained on (Left) and held out test sets from the same distribution (Right). The $Q$ function trained on Coverage+Success demonstrates substantial overfitting in the mismatch between the distribution of $Q$ values between train and test while the one trained on VBT does not.

much more like Success alone since the classification problem between the two datasets is easy enough that the LfP data is completely ignored).

To isolate this issue, we plot histograms of the learned $Q$ values on the train datasets and held out test datasets for both the Coverage+Success and VBT data in Fig. 6. The histograms confirm that the $Q$ function trained on Coverage+Success assigns low values to Coverage transitions from the training set and high values to Success transitions from the training set. But, these $Q$ functions do not generalize. The test set yields a very different distribution of values. In contrast, VBT data resolves the overfitting issue by ensuring that the only visual differences between failure and success are task-relevant, which facilitates better generalization.

# E   Discussion

In this paper, we introduced VBT, a protocol for robot data-collection for offline policy learning. We showed how value functions learned using VBT data tend to be more accurate, leading to robust policy learning with OffRL. We compared VBT to a number of other data collection protocols, and found that policies learned using VBT result in superior performance. We demonstrated these results on a real world, vision-based deformable manipulation task.

While our method significantly outperforms other data-collection protocols, it has certain limitations. VBT requires a teleoperator to understand the relevant failure modes for the task (in our grasping task, this corresponds to narrowly missing the T-shirt), and the ability to recover and re-attempt.

In future work, it would be interesting to apply VBT to tasks with more variety, such as collecting a dataset consisting of interactions with many different objects, instead of just one object. Finally, while we studied a relatively short-horizon task, future work can look into applying VBT to long-horizon tasks.