# OpenReview forum: "Visual Backtracking Teleoperation: A Data Collection Protocol for Offline Image-Based RL"
_NeurIPS.cc/2022/Workshop/Offline_RL — Offline RL Workshop NeurIPS 2022_

### Official Review · Reviewer_J4p7 · 2022-10-18
**Nice empirical evaluation on one domain, but needs more for validating the method**

**Rating:** 6
**Confidence:** 3

**Review:**

The paper presents a protocol for data collection in offline RL that uses trajectories that demonstrate failure, recovery and success. This allows the offline RL method (IQL) to learn high quality value function and subsequently policies.

Strengths:

1. Addresses an important question of data collection for deriving improvements in offline RL.
2. Proposes a simple enough method for data collection requiring ability of teleoperate and does not require reset to arbitrary state.
3. Performs a thorough evaluation for the deformable grasping task.
4. Real robot experiment that validates their data collection protocol.

Weaknesses:

1. Only 1 task seems to be insufficient to verify the claims of benefit from this method.
2. “VBT has the benefit of visually similar features” - It might help to formally define similarity and further understand what kind of data collection prevents overfitting in Q function.

---

### Official Review · Reviewer_NANZ · 2022-10-18

**Rating:** 7
**Confidence:** 4

**Review:**

This paper proposes a protocol for data collection called Visual Backtracking Teleoperation (VBT). The idea is to modify the typical robot teleoperation data collection scheme - instead of having humans collect only successful trajectories, the human teleoperator should demonstrate a failure behavior, a recovery behavior, and finally task success, during each collected trajectory. This would ideally be useful for learning accurate value functions from image observations that would help offline RL learn tasks from small amounts of human data. The method is demonstrated on a T-shirt grasping task on a physical robot.

This paper is clearly relevant to the workshop, and presents an interesting perspective on data collection with humans and how offline RL can benefit, so I recommend acceptance. However, I also had a number of comments and thoughts to improve the manuscript, which I include below:

- I would think it would be even better to combine VBT-data and success-only data to make sure entire optimal traces exist - this might be an interesting comparison.
- Having more data collection baselines would be useful. For example human-in-the-loop (dagger-style) data collection (like references 14, 15, 16, 17 in the paper) - would be a very interesting comparison. Is it better for the human to estimate what constitutes recovery behavior, or for a policy to fall into states that need recovery? It would be useful to compare against typical data collection approaches in the offline RL community as well (for example the D4RL [1] and RL-Unplugged [2] datasets, where replay buffers or agents of random, medium, and expert quality are used, or even the robomimic datasets [3], where human data of mixed quality is used).
- The Q-function evaluations are interesting and useful, but they also seem a little biased towards ensuring VBT will do better - since they are on held-out VBT-style trajectories, which only VBT saw during training.
- Would be nice to see more results in simulation as well, along with released datasets - to more easily reproduce results, compare with established numbers on simulation tasks, and be able to more easily tell the difference between VBT-collected datasets and success-only datasets. It would also make the argument for VBT data collection more convincing - right now there is only 1 task shown directly on a real robot.
- Similarly it could be useful to show experiments with multiple human operators, to show that this paradigm is generally useful and independent of human supervisor.
- Table 1: IQL is only 6% better than BC on VBT data, and BC is tied with AWAC. This makes it seem like the offline RL part is not very useful, and rather the data quality and recovery behaviors matter more. More analysis here would be great.
- Typo: Figure 2: "trainde"


References:

[1] https://arxiv.org/abs/2004.07219

[2] https://arxiv.org/abs/2006.13888

[3] https://arxiv.org/abs/2108.03298